# Interactive Effects of Salicylic Acid and Nitric Oxide in Enhancing Rice Tolerance to Cadmium Stress

**DOI:** 10.3390/ijms20225798

**Published:** 2019-11-18

**Authors:** Mohammad Golam Mostofa, Md. Mezanur Rahman, Md. Mesbah Uddin Ansary, Masayuki Fujita, Lam-Son Phan Tran

**Affiliations:** 1Department of Biochemistry and Molecular Biology, Bangabandhu Sheikh Mujibur Rahman Agricultural University, Gazipur 1706, Bangladesh; mostofa@bsmrau.edu.bd; 2Department of Agroforestry and Environment, Bangabandhu Sheikh Mujibur Rahman Agricultural University, Gazipur 1706, Bangladesh; mrahman@bsmrau.edu.bd; 3Department of Biochemistry and Molecular Biology, Jahangirnagar University, Savar, Dhaka 1342, Bangladesh; ansarymu.ju@gmail.com; 4Laboratory of Plant Stress Responses, Department of Applied Biological Science, Faculty of Agriculture, Kagawa University, Kagawa 761-0795, Japan; fujita@ag.kagawa-u.ac.jp; 5Plant Stress Research Group, Ton Duc Thang University, Ho Chi Minh City 700000, Vietnam; 6Faculty of Applied Sciences, Ton Duc Thang University, Ho Chi Minh City 700000, Vietnam

**Keywords:** cadmium toxicity, growth inhibition, oxidative stress, rice tolerance, ROS detoxification, salicylic acid, sodium nitroprusside

## Abstract

Cadmium (Cd) is one of the prominent environmental hazards, affecting plant productivity and posing human health risks worldwide. Although salicylic acid (SA) and nitric oxide (NO) are known to have stress mitigating roles, little was explored on how they work together against Cd-toxicity in rice. This study evaluated the individual and combined effects of SA and sodium nitroprusside (SNP), a precursor of NO, on Cd-stress tolerance in rice. Results revealed that Cd at toxic concentrations caused rice biomass reduction, which was linked to enhanced accumulation of Cd in roots and leaves, reduced photosynthetic pigment contents, and decreased leaf water status. Cd also potentiated its phytotoxicity by triggering reactive oxygen species (ROS) generation and depleting several non-enzymatic and enzymatic components in rice leaves. In contrast, SA and/or SNP supplementation with Cd resulted in growth recovery, as evidenced by greater biomass content, improved leaf water content, and protection of photosynthetic pigments. These signaling molecules were particularly effective in restricting Cd uptake and accumulation, with the highest effect being observed in “SA + SNP + Cd” plants. SA and/or SNP alleviated Cd-induced oxidative damage by reducing ROS accumulation and malondialdehyde production through the maintenance of ascorbate and glutathione levels, and redox status, as well as the better activities of antioxidant enzymes superoxide dismutase, catalase, glutathione *S*-transferase, and monodehydroascorbate reductase. Combined effects of SA and SNP were observed to be more prominent in Cd-stress mitigation than the individual effects of SA followed by that of SNP, suggesting that SA and NO in combination more efficiently boosted physiological and biochemical responses to alleviate Cd-toxicity than either SA or NO alone. This finding signifies a cooperative action of SA and NO in mitigating Cd-induced adverse effects in rice, and perhaps in other crop plants.

## 1. Introduction

Modern-environment and human health are continuously challenged by a plethora of pollutants, which are originated by both anthropogenic and geological activities [1]. Among these pollutants, heavy metals have been evolved as one of the major contaminants, endangering the quality of lives of diverse habitants ranging from microbial communities to plants, animals, and humans [2,3]. In the recent decades, cadmium (Cd) contamination in water and soil bodies has considerably increased because of widespread use of Cd-containing phosphate-fertilizers and pesticides, extensive mining and smelting, and injudicious disposal of industrial sludge and wastes into the environment [2,3]. In Bangladesh, the incidence of Cd contamination in paddy fields adjacent to the industrial zones has been rising at an alarming rate because of the disposal of untreated wastes from drug and garment factories, as well as textile and tannery industry [4].

Although plants have several defense strategies against Cd-toxicity, most of the crop plants display severe vulnerability if they accumulate even a small amount of Cd in their leaf tissues (5–10 µg g^−1^ dry weight) [5]. Upon entering into the cells, Cd interferes with various cellular mechanisms, such as water balance, mineral homeostasis and antioxidant defense, resulting in poor growth and development of Cd-exposed plants [3,6]. Cd has strong affinities for glutathione (GSH) and thiol-containing enzymes; and thus, excessive Cd can lead to the depletion of GSH content, and deactivation of GSH-dependent reactive oxygen species (ROS)-detoxification system [7,8]. Additionally, Cd-mediated inactivation of metal containing antioxidant enzymes, such as catalase (CAT) and superoxide dismutase (SOD), and stimulation of nicotinamide adenine dinucleotide phosphate (NADPH) oxidases indirectly contribute to the accumulation of ROS in cellular compartments, such as apoplast, chloroplast, mitochondria, and peroxisomes [6,9,10]. Despite having cellular signaling roles at low concentrations, excessive ROS are highly reactive, and display differential degrees of toxicities to proteins, membrane lipids, and nucleic acids, leading to generation of oxidative stress [11]. To protect the cells from the destructive effects of ROS, plants enhance their antioxidant defense system, which comprises of enzymes like SOD, CAT, ascorbate peroxidase (APX), glutathione peroxidase (GPX), and glutathione *S*-transferase (GST), and non-enzymatic metabolites like ascorbate (AsA) and GSH [11]. Like ROS, methylglyoxal (MG), a highly reactive 2-oxo-aldehyde, is also produced in plant tissues under the conditions of Cd-stress [12]. In addition to its direct effects on cellular components, MG can induce oxidative stress by generating ROS and forming glycation end products with proteins and DNA [12]. In response to MG accumulation, a cellular glyoxalase (Gly) system is activated to detoxify MG with the assistance of Gly I and II enzymes, and cofactor GSH. However, the ROS and MG-detoxifying defense mechanisms are only effective up to a certain level of Cd-toxicity [8]; therefore, plants need additional supports from their growers to boost their defense strategies to reduce excessive Cd-induced adverse effects. One of the supports is to supply exogenous compounds, including signaling molecules like salicylic acid (SA) and nitric oxide (NO), which are known to modulate plant defense mechanisms in order to withstand deleterious effects of various abiotic stresses, including heavy metal toxicity [13,14]. 

The modulation of the endogenous SA levels by exogenous addition of SA enhances seed germination, photosynthesis, thermogenesis, transpiration, and membrane formation under various stress condition [15,16,17]. SA also stimulates the expression of defense-associated genes, including *StSABP2*, *StSOD,* and *StAPX* to enhance potato (*Solanum tuberosum*) acclimatization under Cd-stress conditions [17,18]. Furthermore, the beneficial output of SA functions pertains to its interaction with other phytohormones and signaling molecules. For instances, crosstalk of SA with NO reveals activation of antioxidant capacity and photosynthetic adaptability of plants in responses to environmental assaults like drought and salinity [19,20,21,22]. NO is an endogenous gaseous molecule, which has signaling roles in plant acclimatization and tolerance to abiotic stresses [13,14]. The NO-mediated enhancement of GSH production and activation of antioxidant defense contributed to the alleviation of oxidative stress induced by heavy metal toxicity in many plant species, such as rice (*Oryza sativa*), wheat (*Triticum aestivum*) and mustard (*Brassica juncea*) [8,13,23,24]. Furthermore, combined applications of SA and sodium nitroprusside (SNP), a source of NO, successfully diminished the oxidative injuries induced by osmotic and chilling stresses in wheat [25,26], suggesting their interactive roles in mitigation of oxidative stress triggered by a wide range of environmental constraints. 

Crop plants cultivating in Cd-contaminated soils often show high Cd accumulation in edible parts in addition to poor growth performance [27,28]. Numerous reports have demonstrated that cereal crops, especially rice, are responsible for oral intake of Cd, which eventually induces serious hazardous effects on human health, as Cd has mutagenic and carcinogenic effects on various organs, including kidneys, liver, and lungs [1,29]. It has been reported that Bangladesh ranked first among 12 countries under a study by American Chemical Society (ACS) considering the level of Cd in rice samples, and the high traces of toxic Cd (0.01 to 0.3 ppm kg^−1^) have raised serious public health concerns [4]. Therefore, effective strategies should be developed to prevent Cd uptake and accumulation in rice to allow its possible growth and production in Cd-polluted soils for food and feed. Although SA and NO are well-known regulators of heavy metal stress tolerance, their interplay against Cd-stress has not yet been examined in rice at physiological and biochemical levels in a comprehensive manner. Thus, the aim of the current study was to assess the responses of rice seedlings under excessive Cd in the presence or absence of SA and/or NO. For this purpose, the individual and collective effects of SA and NO in mitigating Cd-induced phytotoxic effects in rice were evaluated by assessing plant growth-associated parameters (phenotypes, plant biomass, and photosynthetic pigments), Cd uptake, and accumulation in roots and leaves, leaf water balance, oxidative stress generation, and responses of antioxidant defense and glyoxalase systems. Studying cooperative actions of SA and NO in mitigating Cd-induced adverse effects in rice will allow us to understand the associated mechanisms not only in rice but also in other crops, thereby providing us a solution for cultivation of rice, as well as other crops, on fields contaminated with Cd, and perhaps other heavy metals.

## 2. Results

### 2.1. Effects of SA and/or SNP Treatments on Phenotypic Appearance, Biomass, and Cd Homeostasis 

Exposure of rice seedlings to the aqueous solution of Cd (500 µM CdCl_2_) for three (3) days triggered severe phenotypic disturbances, including stunted growth, leaf rolling, burning of leaf tips, chlorosis, and even yellowing of the entire plant (Figure 1A). Conversely, in comparison with “Cd” plants, external supply of exogenous SA, SNP, or SA + SNP reverted the Cd-induced phytotoxic effects by reviving green color, decreasing leaf rolling, and improving the overall phenotypic appearance of treated rice plants. The recovery was more pronounced in “SA + SNP + Cd” and “SA + Cd” plants than in “SNP + Cd” plants (Figure 1A). Cd-stress significantly decreased plant biomass, as observed in dry weight (DW) reduction in “Cd” plants by 15.31% when compared with the untreated plants (Figure 1B). The supplementation of SA, SNP, or SA + SNP diminished the negative effects of Cd-stress on rice growth, as DW was recovered by 15.40, 12.05, and 16.52% in “SA + Cd,” “SNP + Cd,” and “SA + SNP + Cd” plants, respectively, when compared with that of “Cd” plants (Figure 1B). In comparison with control plants, Cd content was elevated by 23847.51% in roots and by 7745.56% in leaves of “Cd” plants (Figure 1C,D). On the other hand, adding SA, SNP, or SA + SNP to Cd solution reduced the Cd contents by 29.27, 24.04, and 42.70% in roots, and by 33.96, 22.54, and 38.24% in leaves of “SA + Cd,” “SNP + Cd,” and “SA + SNP + Cd” plants, respectively (Figure 1C,D).

### 2.2. Effects of SA and/or SNP Treatments on Photosynthetic Pigment Contents, Water Status, and Proline (Pro) Accumulation 

In relation to control plants, a sharp decline in the contents of chlorophyll (Chl) *a* (26.73%), Chl *b* (61.00%), total Chls (34.16%), and carotenoids (18.82%) was observed in the leaves of “Cd” plants (Table 1). In contrast, SA, SNP, or SA + SNP supplementation protected the photosynthetic pigments from Cd-induced toxic effects by enhancing the contents of Chl *a* (6.64, 0.79, and 11.54%), Chl *b* (78.95, 67.59, and 148.95%), total Chls (15.93, 9.36, and 29.19%) and carotenoids (16.41, 1.29, and 20.27%) in the leaves of “SA + Cd,” “SNP + Cd,” and “SA + SNP + Cd” plants, respectively (Table 1). Furthermore, the SNP-mediated restoration on the loss of photosynthetic pigments under Cd-stress was lower than that observed with SA or SA + SNP treatment (Table 1).

Rice plants exposed to Cd-stress displayed attenuation of leaf relative water content (RWC) by 27.23% in “Cd” plants, compared with that of control plants (Table 1). Conversely, Cd-stressed rice seedlings treated with exogenous SA or SA + SNP impressively recovered the RWC by 34.98 and 32.67% in “SA + Cd” and “SA + SNP + Cd” plants, respectively (Table 1). Nevertheless, SNP did not exert significant restoration of the RWC (Table 1). Exposure of rice seedlings to Cd-stress exhibited an enhancement in the content of Pro by 489.94% in the leaves of “Cd” plants relative to that of control plants (Table 1). The supplementation of SA, SNP, or SNP + SA brought about a reduction in the accumulations of Pro by 70.51, 18.66, and 63.14% in the leaves of “SA + Cd,” “SNP + Cd,” and “SA + SNP + Cd” plants, respectively, in comparison with “Cd” plants (Table 1).

### 2.3. Effects of SA and/or SNP Treatments on ROS Accumulation, and the Levels of Hydrogen Peroxide (H_2_O_2_), Lipoxygenase (LOX) activity, and Malondialdehyde (MDA) 

Cd-exposed rice seedlings exhibited an enhanced accumulation of ROS products, including superoxide (O_2_^•−^) and H_2_O_2_, as evidenced by more scattered dark blue spots for O_2_^•−^ (Figure 2A), and more dark brown polymerization patches for H_2_O_2_ (Figure 2B) in the second leaf blades of rice plants, when compared with control plants. The application of SA, SNP, or both SA and SNP displayed lower accumulations of O_2_^•−^ and H_2_O_2_ in the leaves of “SA + Cd,” “SNP + Cd,” and “SA + SNP + Cd” plants (Figure 2A,B). The leaves of rice plants subjected to Cd-stress significantly enhanced the levels of H_2_O_2_, LOX activity, and MDA by 45.73, 73.28, and 311.55%, respectively, as compared with that of untreated control plants (Figure 2C–E). On the other hand, SA, SNP, or SA + SNP application reduced the Cd-induced oxidative stress, as evident by the observed reductions in the levels of H_2_O_2_ (23.66, 20.19, and 24.55%), LOX activity (12.28, 13.29, and 8.70%), and MDA (38.85, 25.11, and 41.14%) in “SA + Cd,” “SNP + Cd,” and “SA + SNP + Cd” plants, respectively, in comparison with “Cd” plants (Figure 2C–E). 

### 2.4. Effects of SA and/or SNP Treatments on the Levels of Ascorbic Acid (AsA), Dehydroascorbate (DHA), glutathione (GSH) and Oxidized Glutathione (GSSG), and Redox Status 

Rice plants subjected to Cd-stress showed a decrease in AsA (35.77%), a significant accumulation of DHA (68.41%), and an attenuation in leaf AsA/DHA ratio (61.83%) when compared with control plants (Table 2). On the other hand, leaves of SA-, SNP-, or SA + SNP-treated rice plant showed enhancements in the AsA levels (25.18, 21.56, and 24.64%), decreases in the DHA contents (37.19, 38.75 and 67.90%), and increases in the AsA/DHA ratios (99.07, 98.26, and 288.43%) in “SA + Cd,” “SNP + Cd,” and “SA + SNP + Cd” plants, respectively (Table 2).

In addition, Cd-stressed rice plants exhibited a decrease in the levels of GSH (37.08%) and GSSG (69.75%), and increase in GSH/GSSG ratio (107.22%) in comparison with control plants (Table 2). Regarding GSH and GSSG under SA-, SNP-, or SA + SNP-supplementation, GSH content increased by 17.53, 11.77, and 130.69%, whereas GSSG content reduced by 9.36, 49.75, 26.60% in the leaves of “SA + Cd,” “SNP + Cd,” and “SA + SNP + Cd” plants, respectively (Table 2). As a result, the GSH/GSSG ratio increased 29.97, 123.78, and 214.29% in the leaves of “SA + Cd,” “SNP + Cd,” and “SA + SNP + Cd” plants, respectively (Table 2).

### 2.5. Effect of SA and/or SNP Treatments on the Activities of Antioxidant Enzymes 

In comparison with control plants, “Cd” plants showed an increase of SOD activity by 45.96% and a decrease of CAT activity by 13.69% in leaves (Figure 3A,B). Treatment of rice plants with SA or SA + SNP led to improvement in leaf SOD activity (14.09 and 17.26%), and also in CAT activity (29.87 and 17.25%) in “SA + Cd” and “SA + SNP + Cd” plants, respectively (Figure 3A,B). On the other hand, “SNP + Cd” plants had a reduction of SOD activity by 15.81%, and a non-significant increase of CAT activity by 1.49% (Figure 3A,B). In addition, the activities of AsA-GSH cycle-related enzymes, including APX, monodehydroascorbate reductase (MDHAR), dehydroascorbate reductase (DHAR), and glutathione reductase (GR) increased in “Cd” plants by 112.89, 2.65, 38.21, and 164.33%, respectively (Figure 3C–F).

In comparison with “Cd” plants, the addition of SA, SNP, or SA + SNP reduced the APX activity (0.55, 8.29, and 16.10%) and GR activity (10.98, 13.00, and 14.59%), but increased the MDHAR activity by 49.24, 48.99, and 39.99% in the “SA + Cd,” “SNP + Cd,” and “SA + SNP + Cd” plants, respectively (Figure 3C,D,F). On the other hand, leaf DHAR activity significantly increased by 18.09 and 5.16% in “SA + Cd” and “SA + SNP + Cd” plants, respectively, whereas it showed non-significant change in “SNP + Cd” plants (Figure 3E). Leaf GPX activity was enhanced by 142.55% in “Cd” plants (Figure 3G). However, applying SA, SNP, or SA + SNP significantly reduced the leaf GPX activity by 29.88, 30.21, 42.06% in “SA + Cd,” “SNP + Cd,” and “SA + SNP + Cd” plants, respectively (Figure 3G). In addition, “Cd” plants showed a decrease of leaf GST activity by 65.13%, but the supplementation of SA, SNP, or SA + SNP to “Cd” plants resulted in a significant increase in the activity of GST by 63.65, 144.74, and 190.49% in “SA + Cd,” “SNP + Cd,” and “SA + SNP + Cd” plants, respectively (Figure 3H). The seedlings subjected to Cd-stress showed an increase in Gly I activity by 32.26%, and a decrease in Gly II activity by 55.61%, while the addition of SA, SNP, or SA + SNP did not improve leaf Gly I activity (reduced by 4.52, 12.04, and 18.76%), but increased the Gly II activity by 29.11, 11.01, and 66.62% in “SA + Cd,” “SNP + Cd,” and “SA + SNP + Cd” plants, respectively (Figure 3I,J).

## 3. Discussion

Since cadmium (Cd) is not essential for plant growth and development, plants do not have specific mechanisms for up-taking Cd by their cells. Instead, Cd is accumulated in plant tissues through the transporters that are involved in the uptake of other essential elements like magnesium, calcium, manganese, iron, and zinc [30]. Once accumulated at excessive concentrations, Cd shows toxic effects by disturbing metabolism of plants, resulting in growth inhibition and biomass reduction [3,31]. In the current study, Cd-stressed rice plants displayed phenotypic changes, including decreased growth, leaf rolling, burning of leaf tips, chlorosis, and yellowing of the entire plant (Figure 1A), suggesting that the presence of excessive Cd can induce toxicity effects on hydroponically grown rice plants. The decrease in biomass of Cd-stressed rice plants (Figure 1B) was also detected in other plant species (e.g., wheat, mustard, *Brassica oleracea,* and *Pisum sativum*) exposed to various levels of Cd for short or long periods [13,31,32,33]. In contrast, addition of exogenous salicylic acid (SA) and/or sodium nitroprusside (SNP) to growth medium was found to mitigate Cd-induced phytotoxic effects by restoring plant growth and biomass (Figure 1B), reviving green color, reducing leaf rolling, and improving the overall phenotypic appearance (Figure 1A). Under the experimental conditions, the Cd-stress mitigating effects of these exogenous chemicals on rice followed the SA + SNP > SA > SNP trend, indicating that the co-application of SA and SNP was the most effective in eliciting rice responses against Cd-induced toxic effects than that of any single signaling molecule. Gondor et al. [34] and Per et al. [13] observed that SA [34] and SNP [13] positively regulate Cd-tolerance in maize (*Zea mays*) and mustard, respectively, demonstrating that application of these growth regulators could be a viable option in counteracting the toxic effects of Cd in important cash crops like maize and mustard. Similarly, co-application of SA and SNP promoted growth of wheat under osmotic stress conditions [26], pointing toward that cooperative actions of SA and NO were effective in mitigation of osmotic stress-mediated toxic effects on soybean performance. 

Hyperaccumulators, which are commonly used for phytoremediation can accumulate a large amount of toxic metals like Cd in their tissues without showing visible toxicity symptoms [35]. Unlike hyperaccumulators, the growth performance of other plant species is linked to their resistance to Cd uptake and accumulation in different parts of the plants [8,32,33]. The present study showed that root- and leaf-Cd contents in Cd-stressed rice plants followed differential patterns with root tissues accumulating much higher amount of Cd (Figure 1C,D). Probably, since the roots were in direct contact with Cd-containing nutrient solution, Cd primarily accumulated in roots, and rice plants therefore employed a resistance strategy to confine its accumulation in the roots with reduced transportation to the aerial parts (Figure 1C,D). Nevertheless, the levels of accumulated Cd in both roots and leaves of Cd-stressed plants likely contributed to their poor growth performance, as reflected in their phenotypic aberrations and biomass reduction (Figure 1A,B). Gondor et al. [34] also reported that Cd predominantly accumulated in the roots, and the levels of leaf-Cd were associated with the toxic effects of Cd on maize grown under hydroponic conditions. On the other hand, SA and/or NO played essential roles in Cd-homeostasis by restricting Cd uptake and accumulation in both roots and leaves of Cd-exposed rice plants (Figure 1C,D). These results are in accordance with the findings of Gondor et al. [34] and Kaya et al. [36], who reported the role of exogenous SA and SNP in controlling Cd-homeostasis by inhibiting Cd accumulations in both roots and leaves of maize and wheat. Noticeably, under Cd-stress, the Cd levels in roots and leaves of “SA + SNP”-supplemented plants were significantly lower when compared with SA- or SNP-treated plants (Figure 1C,D). These results indicated that the co-application of SA and SNP was able to better prevent Cd-uptake than that of SA or SNP, thereby enabling rice plants to the most efficiently counteract Cd-toxicity, as reflected in their better growth performance (Figure 1A). 

Excessive Cd has negative effects on photosynthesis by damaging the photosynthetic pigments and electron transport processes, disrupting chloroplast structure and Chl-protein complexes, deactivating enzymes of Chl biosynthesis and Calvin cycle, and disturbing water balance [6,8,37]. Indeed, in the current study, Cd-exposed rice plants exhibited significant losses of Chls and carotenoids together with disturbance in leaf water status (Table 1), which were also reported in other plant species like mustard and maize [13,34,38]. These results indicated that accumulated levels of Cd in leaf tissues interfered with photosynthetic activity by reducing the supply of photosynthetic pigments and water, consequently decreasing growth and biomass of Cd-stressed plants (Figure 1A,B; Table 1). On the other hand, SA, SNP, or SA + SNP reduced Cd-induced losses of photosynthetic pigments and leaf water (Table 1), implying that SA- and/or NO-mediated growth recovery was partly due to their actions on protection of photosynthetic pigments and leaf water contents under Cd-stress. The roles of SA and NO in protecting photosynthetic apparatus and water status have been observed in other abiotic stresses, including salinity and osmotic stress [20,21,26]. Again, in the present study, the SA and SNP combination was more highly effective than SA or SNP alone in protection of Chls, carotenoids and water contents in Cd-stressed rice leaves (Table 1), suggesting that SA and NO, when co-applied, employed cooperative roles in protecting rice plants from Cd-induced toxic effects. 

Plants’ typical physiological response against water deficiency is to accumulate various osmoprotectants, including Pro to overcoming water shortage-associated damage in tissues [39]. In this study, Cd-induced Pro accumulation showed an inverse relationship with leaf RWC, whereas SA and/or SNP addition resisted water loss in leaves without accumulating Pro under Cd-stress conditions (Table 1). Similarly, pretreatment of barley and wheat plants with SA and SNP, respectively, reversed the high levels of Pro induced by Cd-stress [36,40]. In contrast to our findings, SA and NO supplementation increased the Pro accumulation in relation to the improvement of leaf RWC under Cd-stress in potato and tomato (*Solanum lycopersicum)* [41,42]. It seems that SA and/or SNP adjust(s) other osmoprotectants rather than Pro, at least in this study; and thus, restoration of RWC did not require accumulation of Pro. Moreover, the higher level of Pro observed in the leaves of “SNP + Cd” than “SA + Cd” and “SA + SNP + Cd” plants indicated that the “SNP + Cd” plants suffered more toxicity from Cd-stress than “SA + Cd” and “SA + SNP + Cd” plants (Table 1). Thus, the obtained Pro data suggested that the increased Pro was linked to Cd-stress intensity, rather than Cd-stress acclimation as observed in several plant species (e.g., rice, barley and wheat) suffering heavy metal toxicities (e.g., Cu and Cd) [8,36,40].

Being a non-redox metal, Cd does not directly participate in ROS generation. However, when Cd-levels exceed plant tolerance limits, Cd can trigger excessive accumulations of ROS by interfering with several biochemical mechanisms, including replacement of redox-active iron from proteins, depletion of ROS-scavenging enzymatic and non-enzymatic components, and metabolic disturbances during respiration, CO_2_ assimilation, and photorespiration [6,8,10,18]. This study showed that Cd potentiated its phytotoxic effects by generating O_2_^•−^ and H_2_O_2_ in rice leaves (Figure 2A–C). The accumulated levels of ROS showed a positive correlation with the increased levels of MDA in Cd-stressed leaves (Figure 2E), suggesting that Cd-induced ROS led to an enhancement oxidative burst and membrane lipid peroxidation. Additionally, activation of LOX in Cd-exposed leaves also contributed to lipid peroxidation (Figure 2D,E), thereby intensifying the membrane damage in rice leaves under Cd-stress conditions. Kaya et al. [36] also observed that Cd once accumulated at toxic level participated in membrane lipid peroxidation by accumulating excessive ROS. The Cd- accumulation in rice leaves resulted in an inefficient and disturbed antioxidant system, since the levels of AsA and GSH, which are important for several defense mechanisms [11], and their respective redox ratios severely declined in Cd-stressed plants (Table 2). The Cd-mediated depletion of AsA and GSH contents, and redox ratios (AsA/DHA and GSH/GSSG) has also been reported in other plant species, including mustard and Pakchoi (*Brassica chinensis*) [43,44]. In addition, in the present study, SOD and CAT, which together constitute the primary enzymatic defense against ROS by removing O_2_^•−^ and H_2_O_2_ [11], displayed increased and decreased activities, respectively, in Cd-stressed leaves (Figure 3A,B), causing H_2_O_2_ accumulation (Figure 2B,C) that could lead to the formation of hydroxyl radical (^•^OH). The Cd-stressed plants seemed to compensate reduced CAT activity by stimulating the AsA-GSH system through the activation of APX, DHAR, and GR (Figure 3B,C,E,F). However, an unchanged leaf-MDHAR activity led to an inefficient AsA-GSH system for the removal of H_2_O_2_ and maintenance of reduced forms of AsA in Cd-treated plants (Figure 2B and Figure 3D; Table 2). Thus, despite the increased activities of several antioxidant enzymes (e.g., SOD, APX, DHAR and GR), ROS levels in Cd-stressed plants remained significantly high (Figure 3A,C,E,F), indicating that activation of enzymatic system was not high enough to detoxify excessive levels of ROS induced by Cd (Figure 2A–C). Moreover, the declines in AsA and GSH contents and redox balance (e.g., AsA/DHA and GSH/GSSG ratios) (Table 2) in addition to the decreased activities of CAT and GST (Figure 3B,H) enforced generation of severe oxidative stress in Cd-stressed plants. Ahmad et al. [38] also detected high H_2_O_2_ levels in the leaves of Cd-stressed Brassica plants even after increasing the activities of SOD, APX, and GR, suggesting an imbalance in production and removal of H_2_O_2_ under Cd-stress conditions.

However, Cd-induced accumulation of O_2_^•−^, H_2_O_2_, and MDA and the activity of LOX in rice leaves were significantly diminished by supplying exogenous SA and/or SNP to Cd-stressed seedlings (Figure 2A–E), which ascertained an important role of these signaling molecules in mitigating oxidative stress and membrane damage induced by excessive Cd. SA and/or SNP supplementation amended Cd-mediated reduction of AsA and GSH levels, and AsA/DHA and GSH/GSSG ratios (Table 2), which together led to a better antioxidant system for combating Cd-phytotoxicity. Furthermore, SA + SNP combination exhibited better performance in restoration of AsA/DHA and GSH/GSSG ratios, when compared with their individual application, indicating that co-application of SA and NO helped maintain better redox balance for boosting antioxidant system under Cd-stress that their individual application (Table 2). SA and/or SNP was particularly involved in uplifting the activities of SOD and CAT, which coordinately restricted elevation of O_2_^•−^ and H_2_O_2_ in the leaves of Cd-stressed rice plants. These results further indicated that the reduction of H_2_O_2_ in “SA + Cd,” “SNP + Cd,” and “SA + SNP + Cd” plants by the actions of SOD and CAT (Figure 3A,B) did not require a rise in the activities of APX, DHAR, and GR involved in the AsA-GSH cycle (Figure 3C,E,F). However, a remarkable increase in MDHAR activity, which recycles monodehydroascorbate to AsA [11], in “SA + Cd,” “SNP + Cd,” and “SA + SNP + Cd” plants coincided with their increased levels of AsA and AsA/DHA ratio (Table 2). Thus, the antioxidant data obtained from the present study indicated that SA and/or SNP, by intensifying SOD, CAT, and MDHAR activities as well as maintaining the levels of APX, DHAR, and GR activities above the control levels, coordinated the activities of antioxidant enzymes in the presence of enhanced AsA, GSH, and redox status, thereby contributing to the regulation of ROS levels in rice plants grown under Cd-stress (Figure 2A–C, Figure 3A–F; Table 2). Dong et al. [22] also found that the activities of antioxidant enzymes like SOD and CAT were upregulated by co-application of SA and SNP to eliminate ROS induced by salinity in cotton (*Gossypium hirsutum*).

GSH-dependent GPX-GST system and Gly system are important in detoxification of lipid hydroperoxides and reactive aldehydes, thereby conferring tolerance to heavy metal toxicity [11,12]. In this study, Cd-stress caused a significant increase of GPX activity, while GST activity declined (Figure 3G,H), indicating that GPX-GST system did not detoxify the lipid peroxidation-derived products in rice plants exposed to Cd-stress. The depletion of GSH and GSH/GSSG ratio because of Cd-stress might contribute to the reduced function of GST (Figure 3H; Table 2). Although Cd-stress stimulated Gly I activity, it showed a negative effects on Gly II activity (Figure 3I,J), which led to an inefficient Gly system that was unable to protect cellular organelles from MG-toxicity. Furthermore, inefficient recycling of reduced GSH because of the reduced Gly II activity might contribute to the declined level of GSH and decreased GSH/GSSG ratio in Cd-stressed plants (Figure 3J; Table 2). The Cd-mediated imbalance in GPX-GST system, and inactivation of Gly II were also previously reported in rice [8]. On the other hand, addition of SA, SNP, or SA + SNP to Cd-stressed plants reversed the Cd-induced enhancement of GPX activity, but kept the GPX activity significantly higher than that of Cd-stressed alone plants (Figure 3G). Moreover, increased GST activity, particularly in “SA + SNP + Cd” plants indicated that SA and NO might have a cooperative function to promote rice defense against reactive metabolites generated because of excessive Cd accumulation (Figure 1C,D and Figure 3H). Likewise, in case of Gly enzymes, these signaling molecules either individually or in combination did not show effects on Gly I activity, but maintained a higher activity of Gly II in Cd-stressed plants compared with that of Cd-stressed only plants, particularly when they were co-applied (Figure 3I,J). An increased activity of Gly II also endorsed that SA and NO, at least partially, contributed to GSH homeostasis through the Gly system-mediated recycling of GSH (Figure 3J; Table 2). These results together suggest that SA and NO efficiently adjust the GPX-GST and Gly systems to facilitate a greater protection of rice plants from the toxic effects of reactive aldehydes in the presence of excessive Cd. 

## 4. Materials and Methods 

### 4.1. Plant Materials, Growth Conditions, and Treatments

Healthy seeds of rice (*Oryza sativa* cv. BRRI dhan52) were sprouted and hydroponically developed in a growth chamber under controlled conditions (light: 100 µmol photon m^−2^ s^−1^; temperature: 26 ± 2 °C relative humidity: 65–70%) [15]. Commercial hydroponic solution (Hyponex, Japan) was diluted to 5000-fold, and renewed after every three day. Cadmium (Cd) was added to the hydroponic solution as cadmium chloride (CdCl_2_). Fourteen-day-old healthy rice plants were grouped into five treatments, which were (1) control (C), (2) 500 µM CdCl_2_ (Cd), (3) 200 µM SA + 500 µM CdCl_2_ (SA + Cd), (4) 200 µM SNP + 500 µM CdCl_2_ (SNP + Cd), and (5) 200 µM SA + 200 µM SNP + 500 µM CdCl_2_ (SA + SNP + Cd). The plants exposed to the treatment solutions were further grown for a period of three days. The second leaves of the 17-day-old rice plants were harvested for determination of various physiological and biochemical parameters. Three independent replications of each treatment were used. Each replication consisted of 60 rice plants under the same experimental conditions.

### 4.2. Assessment of Plant Biomass, and Quantification of Cd Contents in Leaves and Roots

The biomass of rice plants was assessed by determining the dry weight (DW). To estimate the DW, whole seedlings (n = 10) were harvested from each treatment, and subjected to oven-drying for 48 h at 80 °C, and expressed as g seedling^−1^. Cd contents in the oven-dried rice roots and leaves were quantified following the procedure described by Mostofa et al. [8] with the flame atomic absorption spectrophotometer (Z-5000; Hitachi, Japan).

### 4.3. Photosynthetic Pigment Contents, Relative Water Content, and Proline Content

The levels of chlorophyll (Chl) *a*, *b*, and total Chls, and carotenoids in the freshly harvested second leaves were spectrophotometrically quantified according to the formula reported by Arnon [45], and Lichtenthaler and Wellburn [46], correspondingly. The relative water content (RWC) of the second leaves of rice plants from each treatment was measured based on fresh weight (FW), DW, and turgid weight (TW) according to Mostofa and Fujita [15]. The contents of proline (Pro) in the second leaves of rice plants were quantified following the procedure of Bates et al. [47], and modified by Mostofa and Fujita [15].

### 4.4. Hydrogen Peroxide and Malondialdehyde Levels

Contents of hydrogen peroxide (H_2_O_2_) and lipid peroxidation product malondialdehyde (MDA) in the second leaves of rice plants were quantified following the methods of Mostofa and Fujita [15], and Heath and Packer [48], respectively.

### 4.5. Histochemical Analyses of Superoxide and Hydrogen Peroxide

The technique of Mostofa and Fujita [15] was adopted for the histochemical assays of superoxide (O_2_^•−^) and H_2_O_2_ in the second leaves of rice by using the solutions of 0.1% nitroblue tetrazolium (NBT) and 1% 3, 3’-diaminobenzidine (DAB), respectively. 

### 4.6. Extraction and Estimation of Non-Enzymatic Antioxidants

Freshly harvested second leaves of rice plants (0.5 g/sample) were crushed in 3 mL of extraction buffer (1 mM ethylenediaminetetraacetic acid in 5% ice-cold metaphosphoric acid) and centrifuged at 11,500× *g* for 15 min at 4 °C. The reduced and total ascorbic acid (AsA) contents were quantified following the method described in Mostofa et al. [49]. The contents of total GSH and GSSG (oxidized GSH) were estimated according to the procedure of Griffith [50].

### 4.7. Enzyme Extraction and Assessments of Enzyme Activities 

Extraction buffer (50 mM ice-cold potassium-phosphate buffer, 100 mM potassium chloride, 1 mM AsA, 5 mM *β*-mercaptoethanol and 10% (*v/v*) glycerol) was used for crushing the freshly harvested leaf samples (0.5 g/sample). After centrifugation, the resultant supernatants were collected and kept on ice bath for protein and enzyme analyses. The procedure of Doderer et al. [51] was used to quantify lipoxygenase (LOX, EC 1.13.11.12) activity. Activities of superoxide dismutase (SOD, EC 1.15.1.1) and catalase (CAT, EC 1.11.1.6) were estimated following the procedures of Mostofa and Fujita [15]. Ascorbate peroxidase (APX, EC: 1.11.1.11), monodehydroascorbate reductase (MDHAR, EC 1.6.5.4), and dehydroascorbate reductase (DHAR, EC 1.8.5.1) activities were quantified following the protocols of Mostofa and Fujita [15], whereas the methodology of Mostofa et al. [49] was used for the evaluation of glutathione reductase (GR, EC 1.6.4.2), glutathione *S*-transferase (GST, EC 2.5.1.18), and glutathione peroxidase (GPX, EC: 1.11.1.9) activities. For the determination of glyoxalase (Gly) activities, the methodologies of Hossain et al. [52] and Mostofa and Fujita [15] were followed for Gly I (EC 4.4.1.5) and Gly II (EC 3.1.2.6), respectively.

### 4.8. Determination of Total Soluble Protein Contents

The supernatant extracted for assessing enzyme activity was used for the determination of total soluble protein content following the dye-binding protocol of Bradford [53]. The total soluble protein content of each sample was estimated by developing a standard graph with bovine serum albumin.

### 4.9. Statistical Analysis

The obtained data were subjected to a one-way analysis of variance (ANOVA) to test homogeneity of variance among the data using Statistix 10 software package. Least significant difference (LSD) post hoc test was carried out to identify the significant differences among the treatments at 5% level of significance (*p* < 0.05). 

## 5. Conclusions

The present study revealed physiological and biochemical mechanisms associated with SA- and/or NO-regulated rice tolerance to Cd-stress. The Cd-induced toxic effects on rice growth was expressed in terms of increased Cd accumulations in roots and leaves, phenotypic changes, biomass reduction, leaf water imbalance, oxidative stress generation, and weakening of antioxidant defense and Gly systems. However, simultaneous application of SA and SNP with Cd resulted in substantial improvement of rice responses against Cd-toxicity, leading to improved performance of rice plants under Cd-stress conditions. SA and/or SNP restricted the Cd-uptake and accumulation, protected photosynthetic pigments, maintained leaf water balance, and reduced oxidative damage to membrane through increased activities of the antioxidant defense and Gly system. Although SA and SNP were effective in improving Cd tolerance, their co-operative actions proved to be the most effective in mitigating Cd-induced adverse effects on rice growth performance. Current findings, therefore, suggest that exogenous treatments of rice plants with SA and/or SNP, and perhaps genetic engineering to enhance SA and/or NO pathways in rice plants, could be a viable defense strategy in promoting their growth and development under Cd-stress conditions. 

## Figures and Tables

**Figure 1 ijms-20-05798-f001:**
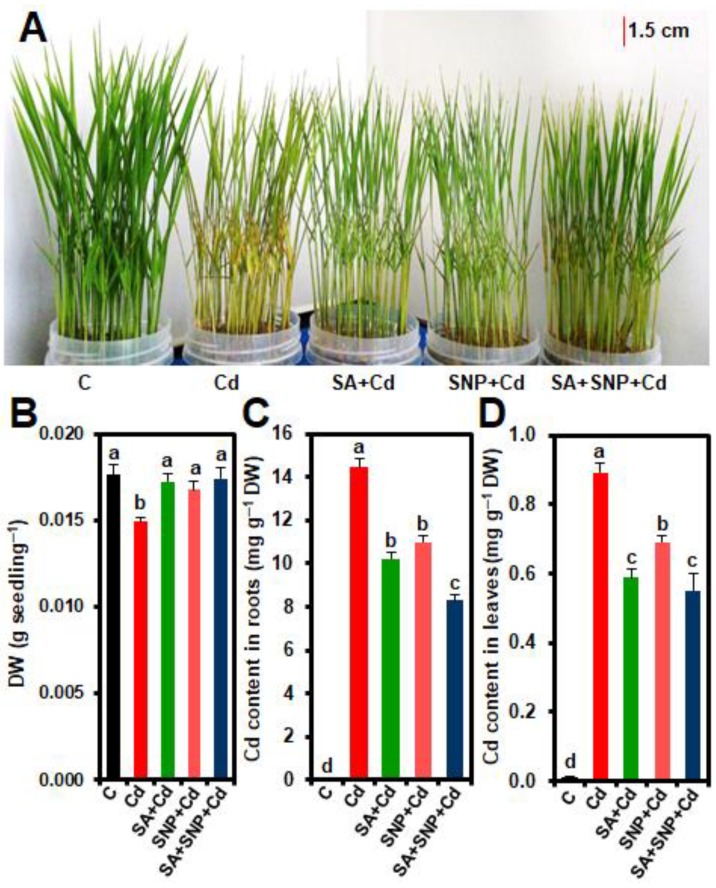
Effects of exogenous salicylic acid (SA), sodium nitroprusside (SNP), or SA + SNP on (**A**) the phenotypic appearance, (**B**) dry weight of rice plants under cadmium (Cd)-stress, (**C**) Cd content in roots, and (**D**) Cd content in leaves. C, Cd, SA + Cd, SNP + Cd, and SA + SNP + Cd correspond to the group of seedlings exposed to only nutrients (control), 500 µM CdCl_2_, 200 µM SA + 500 µM CdCl_2_, 200 µM SNP + 500 µM CdCl_2_, and 200 µM SA + 200 µM SNP + 500 µM CdCl_2_, respectively. Values (means ± SEs) of each treatment were attained from three replications (*n* = 3). Different letters above the bars show statistically significant differences (*p* < 0.05; least significant difference test) among the treatments. DW, dry weight.

**Figure 2 ijms-20-05798-f002:**
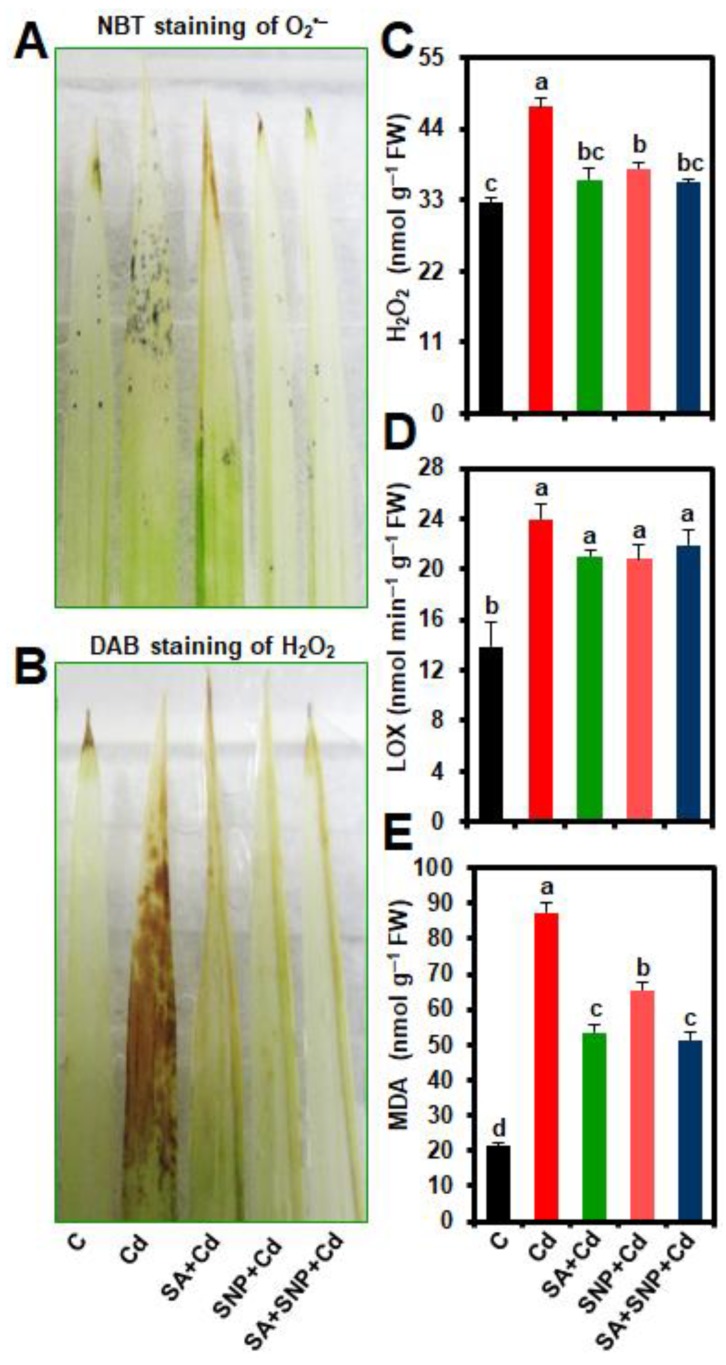
Effects of exogenous salicylic acid (SA), sodium nitroprusside (SNP), or SA + SNP on (**A**) superoxide (O_2_^•−^) accumulation, (**B**) hydrogen peroxide (H_2_O_2_) accumulation, (**C**) H_2_O_2_ contents, (**D**) lipoxygenase (LOX) activity, and (**E**) malondialdehyde (MDA) contents in leaves of rice plants grown with or without cadmium (Cd). C, Cd, SA + Cd, SNP + Cd, and SA + SNP + Cd correspond to the group of seedlings exposed to only nutrients (control), 500 µM CdCl_2_, 200 µM SA + 500 µM CdCl_2_, 200 µM SNP + 500 µM CdCl_2_, and 200 µM SA + 200 µM SNP + 500 µM CdCl_2_, respectively. Values (means ± SEs) of each treatment were attained from three replications (*n* = 3). Different letters above the bars show statistically significant differences (*p* < 0.05; least significant difference test) among the treatments. FW, fresh weight; DAB, diaminobenzidine; NBT, nitroblue tetrazolium.

**Figure 3 ijms-20-05798-f003:**
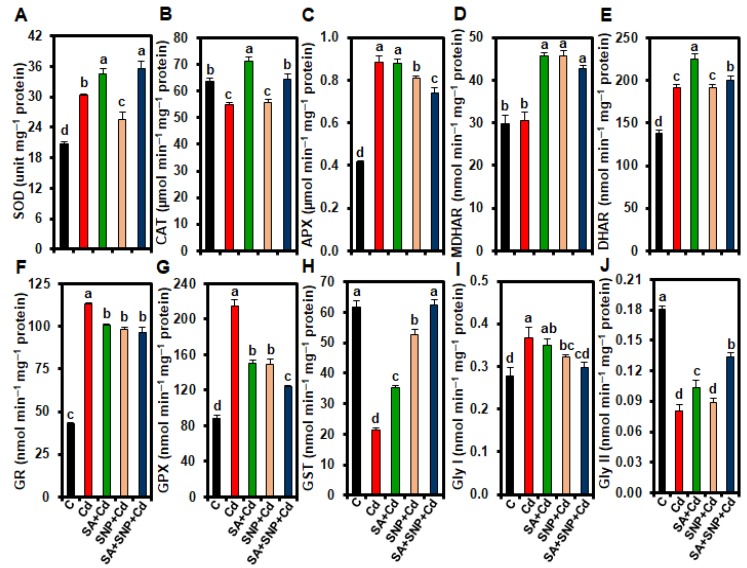
Effects of exogenous salicylic acid (SA), sodium nitroprusside (SNP), or SA + SNP on the activities of (**A**) SOD (superoxide dismutase), (**B**) CAT (catalase), (**C**) APX (ascorbate peroxidase), (**D**) monodehydroascorbate reductase (MDHAR), (**E**) dehydroascorbate reductase (DHAR), (**F**) glutathione reductase (GR), (**G**) GPX (glutathione peroxidase), (**H**) GST (glutathione *S*-transferase), (**I**) glyoxalase (Gly) I, and (**J**) Gly II in the leaves of rice plants with or without cadmium (Cd). C, Cd, SA + Cd, SNP + Cd, and SA + SNP + Cd correspond to the group of seedlings exposed to only nutrients (control), 500 µM CdCl_2_, 200 µM SA + 500 µM CdCl_2_, 200 µM SNP + 500 µM CdCl_2_, and 200 µM SA + 200 µM SNP + 500 µM CdCl_2_, respectively. Values (means ± SEs) of each treatment were attained from three replications (*n* = 3). Different alphabetical letters above the bars show statistically significant differences (*p* < 0.05; least significant difference test) among the treatments.

**Table 1 ijms-20-05798-t001:** Effects of exogenous salicylic acid (SA), sodium nitroprusside (SNP), or SA + SNP on the contents of chlorophyll (Chl) *a*, Chl *b*, total Chls, carotenoids, relative water content (RWC), and proline (Pro) in the leaves of rice plants grown with or without cadmium (Cd).

Treatments	Chl *a* (mg g^−1^ FW)	Chl *b* (mg g^−1^ FW)	Total Chls (mg g^−1^ FW)	Carotenoids (mg g^−1^ FW)	RWC (%)	Pro (µmol g^−1^ FW)
C	2.54 ± 0.07 ^a^	0.70 ± 0.03 ^a^	3.24 ± 0.09 ^a^	0.73 ± 0.00 ^a^	98.78 ± 0.60 ^a^	0.08 ± 0.00 ^e^
Cd	1.86 ± 0.01 ^d^	0.27 ± 0.02 ^c^	2.13 ± 0.01 ^e^	0.59 ± 0.01 ^c^	71.88 ± 3.78 ^b^	0.47 ± 0.01 ^a^
SA + Cd	1.98 ± 0.03 ^b,c^	0.49 ± 0.01 ^b^	2.47 ± 0.04 ^c^	0.69 ± 0.00 ^b^	97.02 ± 0.52 ^a^	0.14 ± 0.01 ^d^
SNP + Cd	1.87 ± 0.01 ^c,d^	0.46 ± 0.02 ^b^	2.33 ± 0.02 ^d^	0.60 ± 0.01 ^c^	75.91 ± 2.83 ^b^	0.38 ± 0.01 ^b^
SA + SNP + Cd	2.07 ± 0.05 ^b^	0.68 ± 0.04 ^a^	2.75 ± 0.02 ^b^	0.71 ± 0.01 ^a^	95.37 ± 1.25 ^a^	0.17 ± 0.01 ^c^

C, Cd, SA + Cd, SNP + Cd, and SA + SNP + Cd correspond to the group of seedlings exposed to only nutrients (control), 500 µM CdCl_2_, 200 µM SA + 500 µM CdCl_2_, 200 µM SNP + 500 µM CdCl_2_, and 200 µM SA + 200 µM SNP + 500 µM CdCl_2_, respectively. Values (means ± SEs) of each treatment were attained from three replications (*n* = 3). Different letters within the columns show statistically significant differences (*p* < 0.05; least significant difference test) among the treatments. FW, fresh weight.

**Table 2 ijms-20-05798-t002:** Effects of exogenous salicylic acid (SA), sodium nitroprusside (SNP), or SA + SNP on the levels of ascorbic acid (AsA), glutathione (GSH), dehydroascorbate (DHA), and oxidized GSH (GSSG), as well as their redox states (AsA/DHA and GSH/GSSG) in rice plants with or without cadmium (Cd).

Treatments	AsA (nmol g^−1^ FW)	DHA (nmol g^−1^ FW)	AsA/DHA Ratio	GSH (nmol g^−1^ FW)	GSSG (nmol g^−1^ FW)	GSH/GSSG Ratio
C	14209.41 ± 157.51 ^a^	977.65 ± 14.20 ^b^	14.54 ± 0.35 ^b^	425.24 ± 18.25 ^b^	120.63 ± 3.18 ^a^	3.54 ± 0.25 ^e^
Cd	9127.06 ± 115.11 ^d^	1646.47 ± 49.19 ^a^	5.55 ± 0.14 ^d^	267.55 ± 11.30 ^d^	36.50 ± 0.36 ^b^	7.33 ± 0.31 ^d^
SA + Cd	11425.59 ± 59.92 ^b^	1034.12 ± 9.34 ^b^	11.05 ± 0.10 ^c^	314.44 ± 7.22 ^c^	33.08 ± 1.09 ^b^	9.53 ± 0.42 ^c^
SNP + Cd	11094.71 ± 101.68 ^c^	1008.53 ± 13.75 ^b^	11.01 ± 0.19 ^c^	299.05 ± 7.37 ^c^	18.34 ± 0.93 ^d^	16.41 ± 1.03 ^b^
SA + SNP + Cd	11376.18 ± 220.62 ^b^	528.53 ± 12.82 ^c^	21.56 ± 0.84 ^a^	617.22 ± 9.16 ^a^	26.79 ± 0.48 ^c^	23.04 ± 0.07 ^a^

C, Cd, SA + Cd, SNP + Cd, and SA + SNP + Cd correspond to the group of seedlings exposed to only nutrients (control), 500 µM CdCl_2_, 200 µM SA + 500 µM CdCl_2_, 200 µM SNP + 500 µM CdCl_2_, and 200 µM SA + 200 µM SNP + 500 µM CdCl_2_, respectively. Values (means ± SEs) of each treatment were attained from three replications (*n* = 3). Different letters within the columns show statistically significant differences (*p* < 0.05; least significant difference test) among the treatments. FW, fresh weight.

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
