# Peer review of "Interactive Effects of Salicylic Acid and Nitric Oxide in Enhancing Rice Tolerance to Cadmium Stress"

_ijms, 2019, doi:10.3390/ijms20225798_

Round 1
Reviewer 1 Report
Review of manuscript
The manuscript is very intertesting but lack practice information about detoxification stress of rice
Title
Ok, correct
Key words: I have suggested add salicylic acid and sodium nitroprusside
Introduction
Line 51
In Bangladesh can used industral sewage sludge to environment ?, agicultural land ?
Line 57
I suggest add paper about plant growth in Cd polluted soils. For example:
Bączek-Kwinta R., Juzoń K., Borek M., Antonkiewicz J. 2019. Photosynthetic response of cabbage in cadmium-spiked soil. Photosynthetica, 57, 3, 731-739. DOI: 10.32615/ps.2019.070
Why used plnt – rice in research, in experiment with SA and NO ?
Material and methods
Line 460
Were the Certified reference materials (CRM) used in analysis of Cd
Results
Was the electrolyte leakage from the leaves tested?
Good discibed results
Discussion
Please give information about remediation of Cd by plants, because many plants uptake Cd from soils
For example:
Kusznierewicz B., Bączek-Kwinta R., Bartoszek A., Piekarska A., Huk A., Manikowska A., Antonkiewicz J., Namieśnik J., Konieczka P. 2012. The dose-dependent influence of zinc and cadmium contamination of soil on their uptake and glucosinolate content in white cabbage (Brassica Oleracea var. Capitata F. Alba). Environmental Toxicology and Chemistry, 31, 11, 2482-2489. DOI:10.1002/etc.1977
Conclusion
Why rice cultiveted in polluted soils ?
Author Response
Please see the file attached! Thank you!

Reviewer 2 Report
This research about the cadmium stress in plants is very important on the context of climate change and increase pollution. However, the manuscript needs a major review by authors. The suggestion are given on the manuscript (a file is enclosed).

Author Response
Please see the file attached!

Round 2
Reviewer 1 Report
The manuscript has been improved.
In my opinion the manuscript can accept to print in IJMS
Reviewer 2 Report
The authors presented a very good revision of the manuscript. I have minr changes that authors can correct. I send the file with the corrections to be made by the authors.
